

# A new radiation infrastructure for the Modular Earth Submodel System (MESSy, based on version 2.51)

Simone Dietmüller[1,3], Patrick Jöckel[1], Holger Tost[2], Markus Kunze[3], Cathrin Gellhorn[3], Sabine Brinkop[1], Christine Frömming[1], Michael Ponater[1], Benedikt Steil[4], Axel Lauer[1], and Johannes Hendricks[1]

[1]Deutsches Zentrum für Luft- und Raumfahrt (DLR), Institut für Physik der Atmosphäre, Oberpfaffenhofen, Germany
[2]Johannes-Gutenberg-University Mainz, Institut für Physik der Atmosphäre, Mainz, Germany
[3]Freie Universität Berlin, Institut für Meteorologie, Berlin, Germany
[4]Max-Planck-Institut für Chemie, Abteilung Atmosphärenchemie, Mainz, Germany

*Correspondence to:* S. Dietmüller
(simone.dietmueller@dlr.de)

**Abstract.** The Modular Earth Submodel System (MESSy) provides an interface to couple submodels to a basemodel via a highly flexible data management facility (Jöckel et al., 2010). In the present paper we present the four new radiation related submodels RAD, AEROPT, CLOUDOPT and ORBIT. The submodel RAD (with shortwave radiation scheme RAD_FUBRAD) simulates the radiative transfer, the submodel AEROPT calculates the aerosol optical properties, the submodel CLOUDOPT

calculates the cloud optical properties, and the submodel ORBIT is responsible for Earth orbit calculations. These submodels are coupled via the standard MESSy infrastructure and are largely based on the original radiation scheme of the general circulation model ECHAM5, however, expanded with additional features. These features comprise, among others, user-friendly and flexibly controllable (by namelists) on-line radiative forcing calculations by multiple diagnostic calls of the radiation routines. With this, it is now possible to calculate radiative forcing (instantaneous as well as stratosphere adjusted) of various green-

house gases simultaneously in only one simulation, as well as the radiative forcing of cloud perturbations. Examples of on-line radiative forcing calculations in the ECHAM/MESSy Atmospheric Chemistry (EMAC) model are presented.

## 1 Introduction

The ECHAM/MESSy Atmospheric Chemistry (EMAC) model is a numerical chemistry climate model system that includes submodels describing tropospheric and middle atmosphere processes and their interaction with ocean, land, and human influ-

ences (Jöckel et al., 2006). The Modular Earth Submodel System (MESSy) is used to link different submodels for physical and chemical processes in the atmosphere (Jöckel et al., 2005). With MESSy2 the second development cycle of the Modular Earth Submodel System (see Jöckel et al., 2010) is available. The core atmospheric model of EMAC is the 5th generation of ECHAM general circulation model (Roeckner et al., 2006). One of the fundamental concepts of MESSy is the strict separation of process and diagnostic implementations from the overall technical model infrastructure (e.g., run-control, input/output, memory man-

agement). To achieve this, the model code is organised in 4 different layers (Jöckel et al., 2010): the basemodel layer (BML), the basemodel interface layer (BMIL), the submodel interface layer (SMIL) and the submodel core layer (SMCL). For process





describing submodels, this implies that the code is split into a SMIL module and one or more SMCL modules, at which the SMIL module manages the connections to the overlying standardised model infrastructure, and the SMCL modules contain the actual process descriptions coded independently of the overlying basemodel.

The EMAC radiation submodel RAD4ALL is a re-implementation of the ECHAM5 radiation code, calculating radiative temperature tendencies depending on radiatively active parameters (Jöckel et al., 2006). The input parameters needed for the calculation of the shortwave and longwave radiation fluxes are radiatively active trace gases ($O_3$, $CH_4$, $CO_2$, $N_2O$, CFC$-11$ and CFC$-12$ ), water vapour, cloud cover, clear-sky index, cloud optical properties (shortwave and longwave optical thickness, asymmetry factor and single scattering albedo of cloud particles), aerosol optical properties (shortwave and longwave optical thickness, single scattering albedo and asymmetry factor of aerosols) and orbital parameters (zenith angle of the sun, distance earth-sun and relative day length). The parametrisation of the radiative transfer in the ultraviolet and visible (UV-Vis) and the near infrared (NIR) is based on the 4 band scheme of Fouquart and Bonnel (1980). For the terrestrial (i.e. longwave) part of the spectrum the RRTM (Rapid Radiative Transfer Model, Mlawer et al., 1997) is used, subdividing the longwave spectrum into 16 bands ranging from 3.33 μm–1000 μm. Optionally, the high–resolution shortwave radiation scheme FUBRAD is available within EMAC (Nissen et al., 2007; Kunze et al., 2014) to increase the spectral resolution of the single UV-Vis band in the stratosphere and mesosphere. FUBRAD has an improved spectral resolution of 55 or 106 bands and is therefore especially suited for solar variability studies in the middle atmosphere, where a sufficiently high spectral resolution leads to an improved solar signal in short wave heating rates and thus temperatures (Nissen et al., 2007; Forster et al., 2011). As it operates in the stratosphere and mesosphere, the relevant radiative processes at this altitude are considered, i.e. the heating due to absorption of UV by oxygen and ozone whereas Rayleigh–scattering, and scattering on aerosols and clouds are not considered.

The development of a new EMAC radiation infrastructure was required, as the infrastructure of the radiation submodel RAD4ALL has been associated with many disadvantages:

- In RAD4ALL a multitude of SMIL modules, one for each sub-process, exists.

- The calculation of orbital parameters, aerosol- and cloud optical properties are performed within the radiation submodel RAD4ALL, partly even within in the technically independent SMCL, although these calculations are conceptionally not subject of the radiation calculation itself.

- In RAD4ALL the import of prescribed gridded climatologies of radiatively active gases is directly utilising the data import interface NCREGRID (see Jöckel, 2006).

- A very cryptic, partly confusing code structure makes the implementation of new code, e.g. alternative radiation schemes, or the option of multiple diagnostic calls in one model time step, difficult and error-prone.

Hence, the model advancement described in this paper has been guided by the intention to re-organise RAD4ALL towards a new, more flexible, easily extendable and basemodel independent concept to couple the radiation submodel to the basemodel: only structural changes have been applied, changes in respect to the radiation calculation have not been addressed in this de-



velopment. Hence, identical output to RAD4ALL is achieved with the revised radiation submodel called RAD.

In this paper we present the new modularised EMAC radiation code, which has been derived from RAD4ALL. The new radiation infrastructure, as well as a test case based on it, are presented in Section 2. In the new infrastructure, calculations

of orbital parameters, aerosol optical properties and cloud optical properties are separated from the radiation code resulting in the new independent submodels RAD (including the sub-submodel FUBRAD), ORBIT, AEROPT and CLOUDOPT. Within the submodel RAD online radiative forcing calculations are now possible, representing an important new diagnostic feature in EMAC. An overview over the online radiative forcing calculation in EMAC and examples of radiative forcing calculations are given in Section 3. A short summary is provided on Section 4.

## 2 New infrastructure for the EMAC radiation code

### 2.1 Submodel RAD

The new submodel RAD now provides a flexible, basemodel independent infrastructure for radiation calculation according to the MESSy standard. Fig. 1 shows the revised structure of RAD and its connection to other submodels. The right side of the diagram displays the relationship of the Fortran95 modules of the SMCL and the SMIL. In the basemodel independent

SMCL the Fortran95 modules RAD_ALBEDO, RAD_LONG and RAD_SHORT (RAD_SHORT_v1 and RAD_SHORT_v2, respectively) are USEd[1] by the radiation SMCL module RAD. Two alternative shortwave radiation schemes are possible: the standard ECHAM5 radiation scheme (RAD_SHORT_v1) and the ECHAM5 radiation scheme modified according to Thomas (2008, RAD_SHORT_v2). The latter includes modifications in the ECHAM5 near infrared routines, as the combination of optical properties of different species are inconsistent (for details see Thomas, 2008). The RAD_SHORT_CMN module contains

definitions and an initialisation subroutine, which are commonly used in RAD_SHORT_v1 and RAD_SHORT_v2, respectively (Fouquart and Bonnel, 1980). If the improved high-resolution shortwave radiation sub-submodel FUBRAD (Nissen et al., 2007; Kunze et al., 2014) is switched on, RAD_FUBRAD is used in addition to RAD_SHORT_CMN from the shortwave calculation (RAD_SHORT_v1 or RAD_SHORT_v2). A detailed description of the sub-submodel RAD_FUBRAD is presented in section 2.2. In the SMIL the modules RAD_E5 and RAD_FUB_E5 are responsible for the data transfer from the

ECHAM5 basemodel and other submodels to RAD and from RAD via RAD_E5 to the basemodel. The calculated radiative temperature tendency provides the temperature feedback ($\Delta T_{feed}$) to the base model (see Fig. 1). The radiative temperature tendencies from multiple diagnostic calls are also available as diagnostic variables ($\Delta T_{diag}$).

The left side of Fig. 1 shows the connections (mainly for RAD input) via the MESSy infrastructure submodel CHANNEL (Jöckel et al., 2010) to other submodels. The RAD input variables are provided by the submodels ORBIT (calculation of

30 orbital parameters), IMPORT (data import from external files, Kerkweg and Jöckel, 2015), AEROPT (calculation of aerosol optical properties), and CLOUDOPT (calculation of cloud optical properties). The input for AEROPT is either provided from

---

[1]Fortran95 syntax



the dynamical aerosol models MADE (Lauer et al., 2007), MADE3 (Kaiser et al., 2014), M7 (Vignati et al., 2004), GMXE (Pringle et al., 2010), or from external data via IMPORT. The input for the submodel CLOUDOPT can be selected from the submodel CLOUD or from offline-data via IMPORT.

The RAD user interface (a specific Fortran95 namelist) allows for a trigger ($\Delta t_{rad}$), which explicitly enables radiation calculation, as radiation is not obligatorily called every model time step, as it is computationally intensive. The corresponding time offset is calculated and provided as channel object to the submodel ORBIT. As ORBIT is called every time step, the orbital parameters are calculated with this time offset and provided as channel objects back to RAD (see Fig. 1).

The submodel RAD is controlled by its namelists, where it is possible to select a wide range of different setups, without re-compiling the code. The supplement of this paper contains a detailed description of the namelist settings of RAD. The main features of the radiation namelist are:

- A logical switch for the FUBRAD shortwave radiation scheme.

- The specification of the radiation time step.

- The possibility to modify the solar constant.

- Logical switches for diagnostically calling the radiation scheme multiple times for each time step. These switches are required for radiative forcing calculations (see details in section 3).

- The choice between the shortwave radiation scheme RAD_SHORT_v1 and RAD_SHORT_v2.

- The selection of 18 input variables (listed in the supplement of this paper), required for the radiation calculation. These input variables are given by channel and channel object selection, for instance, from the channels ORBIT, AEROPT, CLOUDOPT and IMPORT, respectively (see Fig. 1). The radiative relevant input variables can either be provided online (via the submodels ORBIT, AEROPT, CLOUDOPT) or offline (e.g. via IMPORT in case the variables are available on a geographical grid). For greenhouse gases (GHGs), besides external data fields via IMPORT, two other options are possible: constant mixing ratios and mixing ratios decaying with altitude.

- The FUBRAD namelists are included in the radiation namelist file. Here, the solar cycle conditions and the spectral resolution can be set.

## 2.2 Sub-submodel RAD_FUBRAD

To achieve a higher spectral resolution for the UV-Vis band, the sub-submodel RAD_FUBRAD (Nissen et al., 2007; Kunze et al., 2014) is used. It operates in the stratosphere and mesosphere, at pressure levels below 70 hPa. RAD_FUBRAD substitutes the UV-Vis band (250–690 nm) of the RAD shortwave radiation parametrisation by 55, or alternatively, by 106 bands. The scheme is based on the Beer–Lambert law, and includes the calculation of shortwave heating rates from the absorption of UV by $O_2$ at the Lyman-$\alpha$ line (121.5 nm, Chabrillat and Kockarts, 1997), the Schumann-Runge continuum and bands (125.5–205 nm, Strobel, 1978), the calculation of shortwave heating rates from the absorption of UV by $O_2$ and $O_3$ in the Herzberg





continuum (206.2–243.9 nm), and by $O_3$ in the Hartley (243.9–277.8 nm), Huggins (277.8–362.5 nm), and Chappuis bands (407.5–690 nm). Efficiency factors according to Mlynczak and Solomon (1993) are included to account for energy loss due to airglow for the Lyman-$\alpha$ line, the Schumann–Runge continuum, and the Hartley bands. Instead of using Rayleigh–scattering in a two stream approximation, backscattering of the atmosphere and surface is considered, where the albedo at 70 hPa in the

UV–Vis, calculated as the ratio of upward and downward directed flux in the UV-Vis, is used to define the upward directed flux in the Huggins and Chappuis bands within FUBRAD. The coupling to the single UV-Vis band, operating at pressures larger than 70 hPa, is done via a coefficient, representing the fraction of downward directed UV-Vis flux at 70 hPa to the respective flux at ToA. The updated version of RAD_FUBRAD has an increased spectral resolution of the Chappuis band (407.5 – 690 nm) from one band in the original version (Nissen et al., 2007) to 6 or 57 in the new version (Kunze et al., 2014).

The band widths and the corresponding $O_3$ absorption cross sections of the additional Chappuis bands are taken from WMO (1986). With the finer spectral resolutions it is now possible to use the observed solar fluxes within each Chappuis band. In the original version (Nissen et al., 2007) the flux in the Chappuis band is scaled to a lower value, as the band width in FUBRAD is reduced, compared to the original version of the parametrisation by (Shine and Rickaby, 1989).The application of non–scaled fluxes allows to create a consistent UV-Vis flux profile of the two combined parametrisations over the complete vertical model

domain and consistent flux diagnostics at ToA and the surface.

### 2.3   Submodel AEROPT

The submodel AEROPT (AERosol OPTical properties) carries out the calculation of aerosol optical properties, which are required as input values for the radiation scheme and are provided by a coupling of the two submodels via the MESSy CHAN-NEL infrastructure.

AEROPT includes several options to provide these required aerosol optical properties to the radiation scheme, i.e. the aerosol optical thickness per grid cell (the total extinction by scattering and absorption of aerosol particles integrated vertically over each grid box), the single scattering albedo (i.e. the ratio of scattering to absorption by the aerosol) and the asymmetry factor (describing the angular distribution of scattering intensity).

Currently there are three options to provide the above mentioned variables to the radiation scheme:

– The first option is using the aerosol climatology TANRE (Tanre et al., 1984) as in the original radiation code of the ECHAM5 and ECHAM6 models. The TANRE climatology provides aerosol concentrations and related aerosol optical properties per unit mass for 5 different aerosol types, which can be individually turned on or off. The climatology is implemented in the form of spectral coefficients, which are converted to grid point space during the model initialisation. During runtime, the model calculated relative humidity at each grid cell is used in conjunction with the climatological

aerosol concentrations from the climatology to calculate the required parameters for the radiation scheme with the help of simplified functions.



- In the second option, the variables can directly be imported from a file via the MESSy submodel IMPORT. For this, the variables are required on a geographical grid as, for instance, provided by the Chemistry-Climate Model Initiative (CCMI) for stratospheric and volcanic aerosols.

- In the third option the variables can be calculated online with the help of aerosol tracer concentrations (component mass and particle number) and their corresponding size distributions. These data can either be provided by external data sources and using passive tracers or calculated online by microphysical aerosol submodels including gas-aerosol partitioning. In the EMAC system there are several aerosol submodels available such as the modal aerosol models MADE (Lauer et al., 2007), MADE3 (Kaiser et al., 2014), M7 (Vignati et al., 2004) or GMXE (Pringle et al., 2010). The online calculation of the aerosol optical properties is then performed with the help of pre-calculated three-dimensional lookup-tables. The look-up tables provide optical properties of aerosol modes as a function of the real and imaginary part of the refractive index and the Mie size parameter (i.e. aerosol size divided by wavelength, $2\pi r/\lambda$). The lookup-tables are calculated with the radiative transfer model code libradtran (Mayer and Kylling, 2005). Libradtran is been used to perform the required Mie calculations for a given aerosol population. Here, it is assumed that the aerosol population is log-normally distributed with a given modal width ($\sigma$). The radiation scheme then takes the particle number weighted average of the values for extinction cross section, single scattering albedo and asymmetry factor from the look-up table as input for the radiative transfer calculations. During runtime, a set of lookup-tables covering all modal widths used within the aerosol submodel is required. For the longwave spectrum only the extinction value is calculated, as the current radiation scheme requires only this parameter.

Aerosol species explicitly considered are water soluble inorganic ions (WASO), black carbon (BC), organic carbon (OC), sea salt (SS), mineral dust (DU) and aerosol water ($H_2O$). The refractive indices for those aerosol species are extracted from various data sources (most of the data are compiled in the HITRAN2004 database) and include wavelength dependencies. The original references are: WASO (mainly using ammonium sulphate values following (Hess et al., 1998)), BC (Hess et al., 1998), SS (Shettle and Fenn, 1979), $H_2O$ (Hale and Query, 1973), OC (Hess et al. (1998); Sutherland and Khanna (1991); S. Kinne, personal communication), DU (Hess et al. (1998), S. Kinne, personal communication).

The refractive indices for each aerosol mode required as input for the look-up tables are calculated assuming an internal mixture of the aerosol components for the hydrophilic modes. A mean refractive index is calculated for each mode-wavelength combination by averaging the refractive indices of the individual components weighted with their volume contributions. The corresponding Mie size parameters are derived from the median radii of the log-normally distributed modes and the respective wavelengths. The wavelength-dependent particle extinction cross section, single scattering albedo, and asymmetry parameter for each mode are then obtained from the look-up table for the appropriate modal width ($\sigma$). For the hydrophobic modes the same approach can be selected as well as assuming an external mixture which results in an averaging of the optical properties of the individual components. Taking into account the particle number concentrations and the grid box's vertical extension, the extinction cross sections can be converted into aerosol optical thicknesses. The optical thickness of the whole aerosol population in the grid cell is then calculated as the sum over all modes. The mean values of the single scattering albedo and



the asymmetry parameter are obtained by averaging over the modes weighted with their optical thickness.To represent mean radiative properties of the aerosol particles for each radiation band, the extinction, single scattering albedo and the asymmetry factor are determined for fixed representative wavelength values and then mapped onto the corresponding radiation bands using a weighting with the solar spectrum.

This technique of calculating the aerosol optical properties on-line from the simulated aerosol concentrations and look-up tables has been applied by Lauer et al. (2007), Pozzer et al. (2012), Pozzer et al. (2015), de Meij et al. (2012), Tost and Pringle (2012), and Righi et al. (2013, 2015).

The AEROPT submodel can be called several times at each time step with different settings simultaneously, such as, for instance, different lookup-tables, the exclusion of individual aerosol species or with the TANRE aerosol climatology, as the calculation is fully diagnostic. All values which are required for the radiation calculation are provided via the MESSy CHAN-NEL interface. Consequently, the coupling structure of the respective radiation call can be provided with the information of aerosol optical properties which are supposed to be used for the respective radiative transfer calculations.

As mentioned before AEROPT is equipped with the option to collect data from external sources e.g., imported from files via IMPORT, or from alternative aerosol schemes, which provide their own calculation of the respective values required for aerosol-radiation-interactions. In addition, AEROPT can provide the aerosol optical properties required for the calculation of photolysis rates, as e.g. used by the submodel JVAL (Sander et al., 2014). For this purpose scattering, absorption, and asymmetry factor can be calculated at additional wavelengths required by JVAL and provided as channel objects.

Besides the three options of providing optical properties to AEROPT, it is also possible to merge two different data sets for aerosol optical properties in the vertical, e.g. using prognostic tropospheric aerosol values combined with the values provided by CCMI for the stratospheric aerosol for the radiation calculations. The merging of two data sets can be done at a given height or as a linear interpolation in pressure between two reference values. It is also possible to add two data sets, for instance in case of missing volcanic aerosols, the corresponding aerosol optical properties can be provided by an external data source and

combined with the online calculated values for prognostic aerosols. The user settings are controlled via namelists, a detailed description of the namelist settings of AEROPT can be found in the supplement of this paper:

- the information (a counting index and the corresponding filenames of the look-up tables) about the desired lookup-tables used (shortwave and longwave spectrum are handled separately),

- the information about the sets of aerosol radiative properties (e.g., GMXE, M7, MADE, MADE3, TANRE), which
explain how the optical properties are going to be calculated (mixing rules, exclusion for certain species, coupling to required input parameters, etc.).

- the option to read a set of aerosol radiative properties from external sources,

- the feature to merge two different datasets of aerosol radiative properties, as required for the RAD submodel, which can either be read in via the external interface or be calculated by AEROPT (or an alternative submodel for calculating
aerosol optical properties). Additionally, optional weighting factors can be included.





## 2.4 Submodel CLOUDOPT

The optical properties of clouds are now calculated in the EMAC submodel CLOUDOPT. The input variables needed for calculating cloud optical properties are cloud cover, cloud liquid and cloud ice water and cloud nuclei concentration. These optical properties are diagnosed at each band to account for their wavelength dependency. Coefficients for the single scattering albedo, the asymmetry factor and the mass extinction are given for cloud liquid droplets and ice crystals. These coefficients

are provided for 4 bands of the shortwave spectrum and for 16 bands of the longwave spectrum (for details see Roeckner et al., 2006). Calculated cloud optical properties then serve as input for the radiation calculation comprising the shortwave and longwave optical thickness, the asymmetry factor and the single scattering albedo of cloud particles.

The CLOUDOPT namelists (see detailed description in the supplement of this paper) comprises mainly four items:

- The model resolution dependent parameters, such as a correction factor for the asymmetry factor of ice clouds, the cloud

inhomogeneity factors of ice and liquid water, and a parameter to correct the asymmetry factor of ice clouds are set. The corresponding (hard-wired) default values of these parameters can thus be overwritten without re-compilation of the code.

- The channel and channel object names of the required input fields are specified: cloud cover, cloud liquid water, cloud ice and cloud nuclei concentration.

- The effective radii of liquid droplets and/or ice can be calculated internally, or be provided by an external channel object.

- The number of (diagnostic) calls of CLOUDOPT in each time step is selected. The required input (items 3 and 4) is set individually for each call.

The submodel CLOUDOPT was further adapted to enable the separate or cumulative calculation of radiative properties for different cloud coverage and/or perturbations, e.g. the coverage with natural clouds and additional contrail coverage. Further-

more, properties of artificial coverages can be determined, e.g. the additional coverage of ice clouds in only one vertical level with a constant optical depth. This allows for example the evaluation of the performance of the radiation code with respect to a benchmark test, similar to Myhre et al. (2009, see the example benchmark test in section 3.4).

## 2.5 Submodel ORBIT

In the new infrastructure of the EMAC radiation calculation the orbital parameters are separated from the radiation calculation.

They are now calculated in the submodel ORBIT. Orbital parameters are depending on the time of the day and the year. The basic equations used are the Kepler equation for the eccentric anomaly, and Lacaille's formula (see Roeckner et al., 2006).

The radiation submodel RAD now accesses the necessary channel objects of the orbital parameters, including the distance sun-earth, the cosines of the zenith angle and the relative day length. As the radiation is not calculated every time step, ORBIT also receives information from RAD (see Fig. 1), namely the offset for the radiation calculation ($\Delta t_{rad}$).

The ORBIT namelists (see detailed description of these namelists in the supplement of the paper) comprise:





- the selection/setting of the orbital parameters, such as the eccentric anomaly, the inclination, and the longitude of perihelion,

- the possibility to distinguish between two orbit calculations, for either annual cycle or perpetual month experiments, respectively, and

- the channel object containing the radiation calculation offset $\Delta t_{rad}$.

## 2.6 Example application: volcanic heating rates

To show the functionality of the new radiation infrastructure, we show a test case: the eruption of Mt. Pinatubo in June 1991, which injected $SO_2$ into the stratosphere and thus modified the radiative balance by additional radiative heating. For our simulations with the revised EMAC radiation infrastructure, we chose a 90 layer model setup (up to 0.01 hPa, approx. 80 km) with a spectral truncation T42 of the dynamical ECHAM5 core. Interactive chemistry was not simulated, but AEROPT was used to provide two different sets of aerosol optical properties: (1) the standard TANRE climatology (i.e., without additional volcanic aerosol) and, (2) the standard TANRE climatology combined (MERGED) with the offline stratospheric aerosol data as provided by CCMI. Note, that the gasphase of $SO_2$ is not radiatively active in our model. In one simulation, the RAD calculation was performed 4 times every 3rd time step per time step: each aerosol input combined with each shortwave radiation scheme (v1 or v2). The simulation has been performed twice, once without and once with the FUBRAD scheme. The resulting 8 different radiation setups are summarised in Table 1.

Fig. 2 shows the resulting simulated volcanic heating rates (in K/d) for the years 1991 to 1993 resulting from the eruption of Mt. Pinatubo. The volcanic heating rates are given as difference between the heating rates simulated with volcanic aerosol (MERGED) and the heating rate simulated without volcanic aerosol (TANRE). The values are averaged for the tropics, i.e., over 5°N-5°S. As to be expected, the patterns for the different setups are similar (and comparable to those of Stenchikov et al. (1998)), since the aerosol optical properties are prescribed. Nevertheless, differences in the absolute values occur. The peak heating rates are larger for the SW-v1 scheme compared to the SW-v2 scheme in accordance with results from Thomas (2008). The application of FUBRAD also decreases the absolute values, as all effects of scattering are not included in FUBRAD. The simulations including FUBRAD, thus only show the effect of volcanic aerosols on the NIR heating rates.

## 3 Calculation of radiative forcing

### 3.1 Technical implementation of radiative forcing calculation in RAD

A new feature in the radiation submodel RAD is the user-friendly and flexible implementation of the online radiative forcing calculation. It is now possible to determine instantaneous as well as stratosphere adjusted radiative forcing online, i.e., during the model simulation, by multiple calls of the radiation scheme. Instantaneous radiative forcing is defined as the change in the net radiative flux with atmospheric temperatures fixed to unperturbed values. In contrast, the concept of stratosphere adjusted radiative forcing, also known as the fixed dynamic heating concept (Ramanathan and Dickinson, 1979; Fels et al., 1980), allows



stratospheric temperatures to adjust to a new radiative equilibrium, without changes in tropospheric variables and stratospheric dynamics. Since the first IPCC report (Houghton et al., 1990) stratosphere adjusted radiative forcing has been the preferred metric used to quantify and rank the numerous components impacting the global climate.

The technical procedure to determine the stratosphere adjusted radiative forcing within a climate model simulation was introduced by Stuber et al. (2001) to the climate model ECHAM4. A second diagnostic temperature is implemented to calculate

the stratosphere adjusted radiative forcing. The reference atmosphere controlled by the first radiation call is not subject to the perturbations, however, the temperature field of the extra diagnostic radiation call experiences additional radiative heating above the tropopause, with dynamical heating remaining identical to the unperturbed reference atmosphere. In the troposphere, the reference temperature and the perturbated diagnostic temperature are identical. To enable the stratospheric temperature to readjust to the new equilibrium, a spin up period of at least 3 months must be considered (Manabe and Strickler, 1964).

It is easy to enable multiple diagnostic calls of the radiation routine in order to determine radiative forcing, after improving the radiation code structure (see section 2.1). Via namelist selection (for detailed description of the radiation namelist see supplement) radiation routines can be called several times within one simulation. The first call is always the reference call and provides the temperature feedback $\Delta T_{feed}$ (see Fig. 1), the other calls are of diagnostic nature. Either instantaneous or stratosphere adjusted radiative forcing can be selected by a namelist switch. With this setup the radiative forcing of various

GHG and aerosol changes can be calculated simultaneously in one model simulation. GHG perturbations can either be given as constant mixing ratios with or without vertical gradient, or as externally prescribed 3-D distributions, or as online calculated, three dimensional fields. All perturbed values are specified via channel object selection in the radiation namelist (see detailed description in the supplement). Hence, radiative forcing can be calculated without extra simulation.

Radiative forcing can be determined either at ToA or at the tropopause, both possibilities are possible in RAD. However,

the determination of radiative forcing at an annual mean tropopause is the usually preferred metric for comparing the climate impact of different GHG perturbations. The annual mean tropopause is used, as no temperature equilibrium can be archived with variable tropopause height. It is possible to calculate radiative forcing at the tropopause via the submodel VISO (Jöckel et al., 2010), which maps 3-D scalar fields in Eulerian representation on arbitrary horizontal surfaces. Moreover, by providing a reference state from offline (e.g. from a pre-calculated stationary reference simulation), it is also possible with this framework

to perform an analysis of feedback during the course of any climate change simulation by multiple call radiative transfer calculations (Chung and Soden, 2015).

In the following subsections we demonstrate the practical advantage of the extended radiative forcing calculation options by a selection of three show cases.

## 3.2  Example 1: Radiative forcing of CO$_2$ increase

The concept of stratosphere adjusted radiative forcing is well known and well established for the case of CO$_2$ change. Its features and merits are repeated here mainly to set the scene for the more interesting non-CO$_2$ cases. The first example, thus, forms a radiative forcing calculation with EMAC using a CO$_2$ increase of 28.8 ppmv, representing the change of CO$_2$ in 2000 relative to 1980. This CO$_2$ change was calculated by the EMAC hind-cast simulation RC1-base-08. The model setup



of this simulation is described in detail by Jöckel et al. (2015). Table 2 lists global mean values for the instantaneous and

5 stratosphere adjusted radiative forcing, both at the top of the atmosphere (ToA, given in brackets) and at the tropopause, while Fig. 3 illustrates the vertical structure of the longwave, shortwave, and net radiative flux changes induced by the $CO_2$ increase.

The main radiative impact of $CO_2$ occurs in the longwave part of the spectrum, whereas the shortwave forcing component is almost zero at the tropopause (but about 18% of the net at ToA because of near infrared absorption in the middle atmosphere). The stratosphere adjusted net radiative forcing ($0.45 \, \mathrm{Wm^{-2}}$) is by about 7% smaller than the instantaneous net radiative forcing

at the tropopause, qualitatively confirming previous findings. The reason for the dampening is the cooling effect of additional $CO_2$ in the stratosphere, reducing the downward longwave flux into the troposphere. The instantaneous net radiative forcing is considerably smaller at ToA compared to the tropopause for the $CO_2$ case (0.27 and $0.48 \, \mathrm{Wm^{-2}}$, respectively). The effect of stratospheric temperature adjustment is to create a new balance of shortwave and longwave fluxes, leading to vertically constant net radiative flux changes above the tropopause (Fig. 3, bottom). Hence, the stratosphere adjusted net radiative forcing has the

same value at ToA and at the tropopause. Note, however, that this does not hold for the shortwave and longwave components.

Model dependencies in radiative forcing may not only arise from the specific radiative transfer scheme used in a given model (here: EMAC), but also from methodical aspects as discussed by, e.g., Forster et al. (1997). In particular, as mentioned above, in our show cases a fixed tropopause from an EMAC reference simulation is used to define the domain where stratospheric temperature adjustment takes place. The temperature adjustment evolves seasonally dependent in the EMAC calculation procedure

(Forster et al., 1997; Stuber et al., 2001), which may lead to slight deviations from the stratosphere adjustment, that is applied when offline radiative transfer models are used for stratosphere adjusted forcing calculations. A consequence is that the stratosphere adjusted radiative forcing profile above the tropopause is constant only in the annual mean. Yet, as already discussed by Forster et al. (1997), this must not be viewed as a conceptual disadvantage, and the online radiative forcing calculations in a CCM like EMAC may have dedicated advantages for many non-$CO_2$ forcings (see section 3.3).

### 3.3 Example 2: Radiative forcing of an ozone-hole like perturbation

Selecting a radiative forcing definition that provides a meaningful indicator of the expected climate effect of ozone concentration perturbations has been the challenge that lead to establishing stratosphere adjusted radiative forcing as a standard procedure for a long time. This example uses a stratospheric ozone change due to stratospheric ozone destruction evolving between 1980 and 2000, again from the EMAC hind-cast simulation RC1-base-08 (see above). The respective stratospheric ozone change

pattern, shown in Fig. 4 as an annual mean, is in good agreement with observations (see Hassler et al. (2013), their fig. 8 ). However, the seasonal cycle is included in the radiative forcing calculations.

In agreement with previous experience (Ramanathan and Dickinson, 1979; Hansen et al., 1997; Forster and Shine, 1997; Christiansen, 1999) the instantaneous net radiative forcing turns out to be extremely ambiguous for this kind of essentially stratospheric ozone perturbation. It changes sign (see table 2) from ToA ($-0.16 \, \mathrm{Wm^{-2}}$) to the tropopause ($+0.06 \, \mathrm{Wm^{-2}}$), a feature controlled by the shortwave component: Less ozone absorption above the tropopause means an energy gain for the

5 troposphere/surface system but an energy loss for the whole atmosphere. Less shortwave absorption above the tropopause, as occurring in this case, means a cooling and changes the downward longwave radiative flux at the tropopause to an extent that the





net radiative forcing at the tropopause changes sign (Fig. 5), giving a negative (albeit small) value of -0.01 $\mathrm{Wm^{-2}}$. As pointed out by Hansen et al. (1997) the negative net forcing has the correct sign to predict a cooling effect in the troposphere/surface system as a result of ozone depletion. Quantitatively our value is smaller than the estimate of this effect given in the last two

10  IPCC reports (-0.05 $\mathrm{Wm^{-2}}$, Solomon et al. (2007); Stocker et al. (2013)), which are based on ozone loss over the period where ozone depleting substances have increased. Our value is close, however, to the estimate of Stevenson et al. (2013), where simulated ozone changes induced by various effects over a similar period as considered in our model simulation were used for the calculation. Anyway, in the present paper our key point is to underpin the usefulness of having a method at hand, which allows to calculate the stratosphere adjusted forcing at the tropopause online in a CCM.

### 3.4  Example 3: Cloud perturbations

The necessity to calculate radiative forcings for cloud changes may arise in context with direct anthropogenic cloud cover change as induced by contrails or ship tracks. It is particularly useful to perform such calculations on a time step basis within a CCM rather than using monthly mean input for an offline radiative transfer model. For example, Frömming et al. (2011) find a reduction of contrail radiative forcing of about 20%, if time-varying (instead of time-averaged) contrail optical depth is used. Rap et al. (2010) even report a reduction of all-sky contrail radiative forcing of more than 35%, if daily correlation of contrails

and natural clouds is accounted for rather than using time mean cloud and contrail properties. Here, however, to evaluate the performance of the EMAC radiation parameterisation in comparison to other radiative transfer codes with respect to thin ice clouds (similar to aviation induced contrails), we carried out an experiment similar to Myhre et al. (2009). In this benchmark test we add a 1% homogeneous contrail cover in one model level with a contrail top of 11 km. The contrails have a constant optical depth of 0.3, while the other optical properties are similar to those reported by Myhre et al. (2009). The instantaneous

as well as the stratosphere adjusted radiative forcing are calculated at the tropopause and at ToA (see corresponding shortwave, longwave and net forcing in table 2). The global annual mean instantaneous net radiative forcing at ToA is 0.109 $\mathrm{Wm^{-2}}$. This result is at the lower end, but within the range given by Myhre et al. (2009) from 0.097 $\mathrm{Wm^{-2}}$ to 0.190 $\mathrm{Wm^{-2}}$. Note that most of the radiation codes tested by Myhre et al. (2009) are more sophisticated than the one presented here which is implemented in a CCM, where a reasonable compromise between accuracy and resource efficiency is essential. In addition to

the instantaneous radiative forcing at ToA, we determine the radiative forcing at the mean tropopause, which is 0.115 $\mathrm{Wm^{-2}}$ for the net. Furthermore, we calculated the stratosphere adjusted radiative forcing at the ToA, which was found to be 0.113 $\mathrm{Wm^{-2}}$ (net), and thereby only deviates by 4% from the instantaneous radiative forcing.

Fig. 6 shows the geographical distribution of the annual mean net radiative forcing at ToA for 1% homogeneous contrail cover. The spatial pattern is dominated by the distribution of natural clouds. The net radiative forcing of the added contrails is

high, where natural cloud cover is low, e.g. over deserts, and is comparably low, in regions with high natural cloud cover, e.g. over the tropics and mid latitudes. The size of minima and maxima, as well as the spatial pattern of the net radiative forcing looks quite similar to the results presented in the intercomparison study of Myhre et al. (2009). Hence, besides its conceptual advantages over offline radiative transfer model estimates for real contrails, this benchmark tests show the suitability of the submodel RAD to estimate the radiative effects of thin ice clouds.





## 4 Summary

The submodel RAD (including the shortwave radiation scheme RAD_FUBRAD) provides a flexible and basemodel independent infrastructure of the radiation transfer calculation according to the MESSy standard. With the new submodels AEROPT, CLOUDOPT and ORBIT the calculations of aerosol and cloud optical properties, as well as the calculation of orbital parameters are now performed within these independent submodels, after having them outsourced from the previous radiation code RAD4ALL. All these new submodels are coupled via the standard MESSy infrastructure to RAD (see Fig. 1).

In the new radiation infrastructure online or offline variables needed for the radiation calculation are selected via namelists. Offline input as e.g. climatologies of radiatively active gases, are now read via the submodel IMPORT, instead of importing them within the radiation code. Thus, the submodel RAD can be applied to easily define different radiation setups with almost arbitrary input. Multiple diagnostic calls of the radiation routine are possible in RAD and, as by-product, radiative forcing can be calculated during the model simulation.

Shown example applications of the now implemented radiative forcing calculations indicate the spectrum of radiative forcing calculations within EMAC.

## 5 Code and data availability

The Modular Earth Submodel System (MESSy) is continuously further developed and applied by a consortium of institutions. The usage of MESSy and access to the source code is licenced to all affiliates of institutions, which are members of the MESSy Consortium. Institutions can be a member of the MESSy Consortium by signing the MESSy Memorandum of Understanding. More information can be found on the MESSy Consortium Website (http://www.messy-interface.org). The developments presented here will be part of the next official release of MESSy.

*Acknowledgements.* We thank Tobias Zinner for providing the radiative transfer model libradtran, which was used to prepare the lookuptables of the Mie size parameters. This work was partly supported by the German Research Foundation (DFG) Research Unit FOR 1095 "Stratospheric Change and its Role for Climate Prediction" (SHARP) and the DLR-project „Verkehrsentwicklung und Umwelt" (VEU). The model simulation RC1-base-08 used for radiative forcing calculations has been performed at the German Climate Computing Centre DKRZ through support from the Bundesministerium für Bildung und Forschung (BMBF).



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





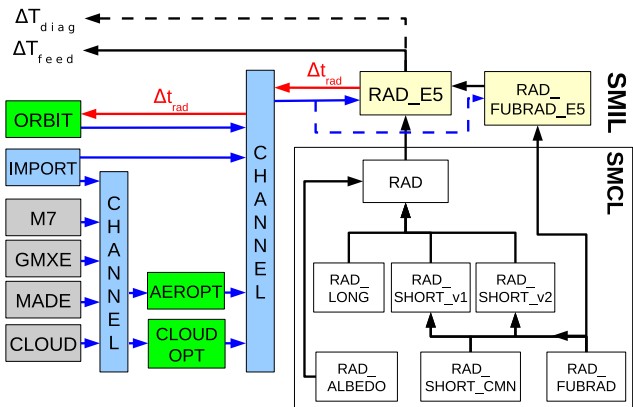

**Figure 1.** Diagram of the revised radiation structure in EMAC. The relationship between the various Fortran95 modules of RAD is given on the right hand side. The different MESSy layers SMCL and SMIL are indicated. The left hand side shows the connection of RAD to other submodels. The grey boxes indicate existing submodels delivering input for the radiation, whereas the green boxes show new submodels, which are now separated from the radiation code. A detailed description is provided in the text.

Thomas, M.: Simulation of the climate impact of Mt. Pinatubo eruption using ECHAM5, Reports on Earth System Science 52, Max Planck
515    Institute for Meteorology, Hamburg, 2008.

Tost, H. and Pringle, K.: Improvements of organic aerosol representations and their effects in large-scale atmospheric models, Atmos. Chem. Phys., **12**, 8687–8709, doi:10.5194/acp-12-8687-2012, 2012.

Vignati, E., Wilson, J., and Stier, P.: M7: An efficient size-resolved aerosol microphysics module for large-scale aerosol transport models, J. Geophys. Res, 109, doi:10.1029/2003JD004485, 2004.

520  WMO: Atmospheric ozone 1985, vol. 1, Global Ozone Res. Monit. Proj. Rep. No. 16, 505 pp., Geneva, Switzerland, 1986.



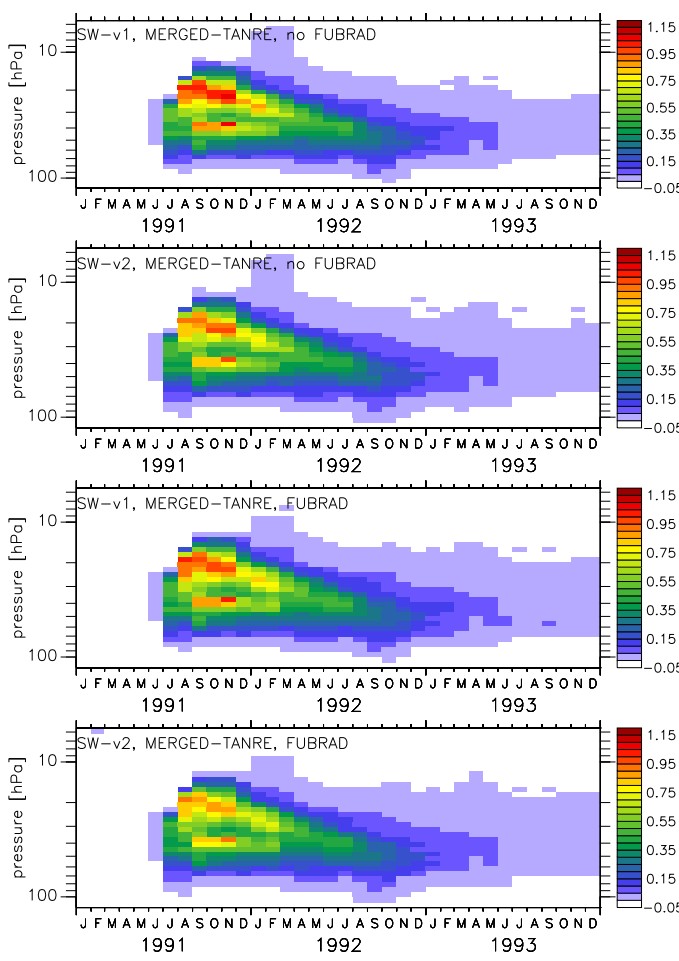

**Figure 2.** Simulated temporal evolution versus pressure altitude of the volcanic heating rates (in K/day) in the tropics (5°S-5°N) due to the eruption of Mt. Pinatubo in June 1991. The different panels show the results for v1 and v2 of the short-wave (SW) scheme, both with and without FUBRAD (as indicated).





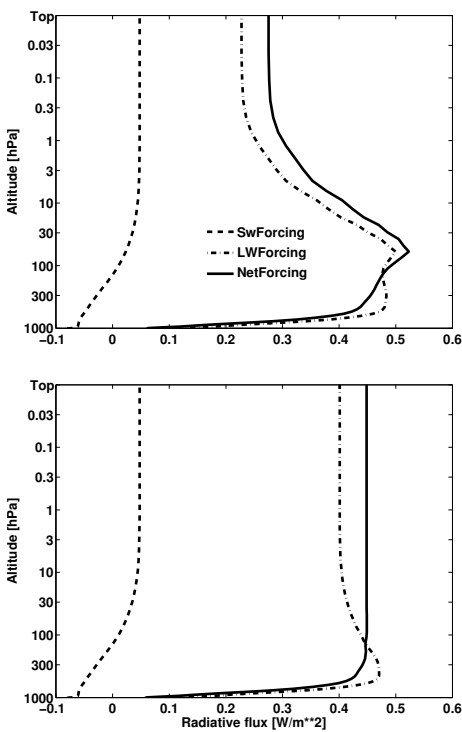

**Figure 3.** Vertical profile of the global and annual mean net, shortwave and longwave instantaneous radiative flux change (top) and stratosphere adjusted radiative flux change (bottom) in $\mathrm{Wm}^{-2}$ resulting from $CO_2$ change between 1980 and 2000.

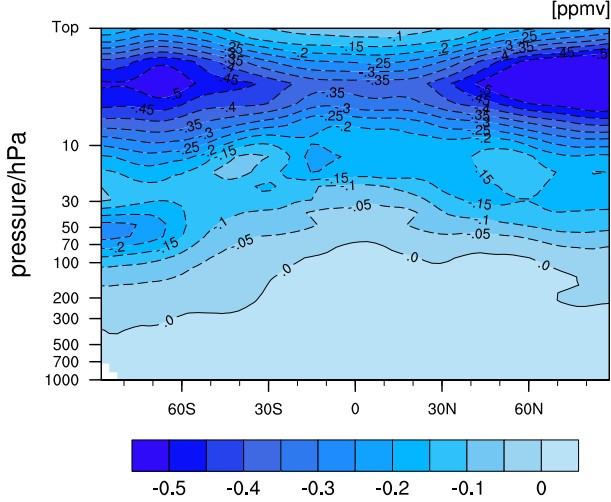

**Figure 4.** Zonal geographical distribution of the annual mean stratospheric $O_3$ change between 1980 and 2000.





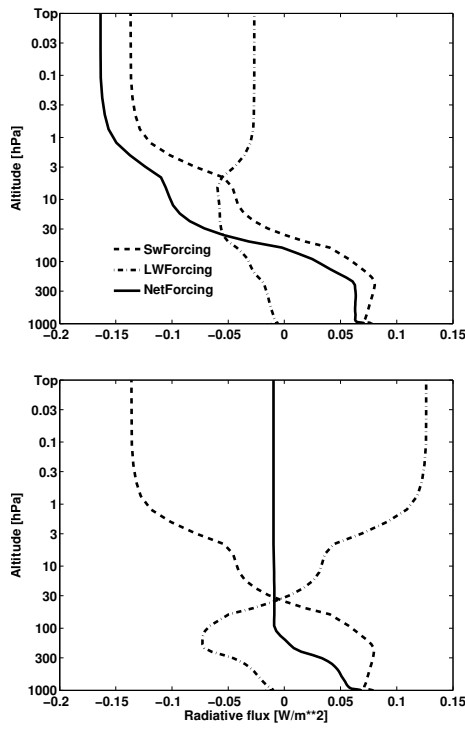

**Figure 5.** Vertical profile of the global and annual mean net, shortwave and longwave instantaneous radiative flux change (top) and stratosphere adjusted radiative flux change (bottom) in $\mathrm{Wm}^{-2}$ resulting from stratospheric $O_3$ change between 1980 and 2000.

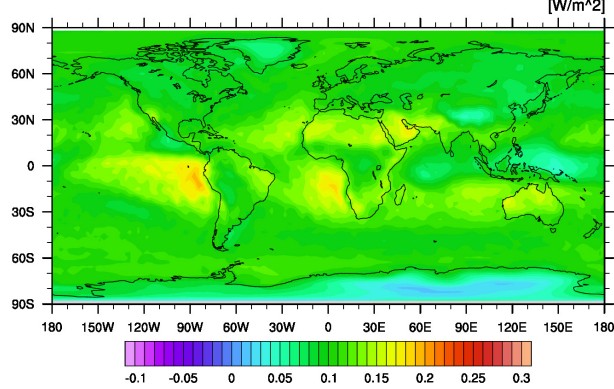

**Figure 6.** Geographical distribution of the annual mean net radiative forcing at ToA for a homogeneous 1% contrail cover.



**Table 1.** Radiation setups (modified heating rates due to the eruption of Mt. Pinatubo) used for testing the new radiation infrastructure. Both simulations cover the years 1991-1993. Variated parameters are the shortwave scheme (v1 or v2), the selection of the FUBRAD radiation scheme, and the selected aerosol input to AEROPT (TANRE or MERGED).

| simulations | sw scheme | FUBRAD | aerosol |
|---|---|---|---|
| 1 | v1 | yes | TANRE |
| 1 | v2 | yes | TANRE |
| 1 | v1 | yes | MERGED |
| 1 | v2 | yes | MERGED |
| 2 | v1 | no | TANRE |
| 2 | v2 | no | TANRE |
| 2 | v1 | no | MERGED |
| 2 | v2 | no | MERGED |

**Table 2.** Annually and globally averaged shortwave (sw), longwave (lw) and net instantaneous and stratosphere adjusted radiative forcing at the tropopause due to changes in $CO_2$ and stratospheric $O_3$ between 1980 and 2000 and due to additional homogeneous 1% contrail cover. The respective values of the radiative forcing at ToA are given in parentheses.

| | instantaneous RF | | | adjusted RF | | |
|---|---|---|---|---|---|---|
| | $CO_2$ | strat. $O_3$ | contrail | $CO_2$ | strat. $O_3$ | contrail |
| sw | 0.002 (0.05) | 0.08 (-0.13) | -0.088 (-0.086) | 0.002 (0.05) | 0.08 (-0.14) | -0.088 (-0.086) |
| lw | 0.48 (0.23) | -0.02 (-0.03) | 0.203 (0.195) | 0.45 (0.40) | -0.09 (0.13) | 0.201 (0.199) |
| net | 0.48 (0.27) | 0.06 (-0.16) | 0.115 (0.109) | 0.45 (0.45) | -0.01 (-0.01) | 0.113(0.113) |