# Peer review of "A new radiation infrastructure for the Modular Earth Submodel System (MESSy, based on version 2.51)"

_Geoscientific Model Development, 2015_

## Referee Comment (RC1) · Anonymous Referee #1 · 16 Mar 2016

This well-written paper describes in a clear and concise fashion how the radiative transfer modules are incorporated into ECHAM5 via the Modular Earth Submodel System (MESSy). Although the paper does not present any new science as such it makes a significant contribution to modelling science in that it provides a very clear description on how to incorporate new modules into a climate model.

Therefore I would recommend that this paper is published subject to some very minor reviews, listed below.

[Figure]

**1   Abstract**

1. Could the author specify that the ECHAM general circulation model (GCM) has been developed by the 'Max-Planck Institut fuer Meteorologie' for clarity purposes.

**2   Introduction**

1. The authors are mentioning that the long-wave spectrum is divided into 16 bands ranging from 3.33-1000 microns. Could the authors please provide the same information for the UV-Vis and NIR. In section 2.2 it mentions that the UV-Vis band ranges from 250nm-690nm and therefore I would assume that the NIR is covered by three bands which range from 0.69micron to 3.3 microns. But I am not sure that this is the case. This needs to be made clearer.

2. The paper mentions that Rayleigh-Scattering is not considered in the submodel RAD_FUBRAD which looks at the UV-Vis regions at pressure levels below 70hPA, i.e. in the stratosphere and mesosphere. It seems to me that in the stratosphere and mesosphere Rayleigh-Scattering would be one of the most important radiative mechanism. Do the authors mean that the short-wave heating rates are not affected much by Rayleigh-Scattering:? or do the submodels RAD and RAD_FUBRAD overlap in presure levels so that stratospheric and mesospheric Rayleigh-Scattering is treated in RAD. This needs to be clarified.

3. The paper also states that the submodel RAD_FUBRAD does not consider scattering by aerosols and clouds, although stratospheric aerosols are know to have an important radiative effect. Again, do the authors mean that they do not have a large effect on the heating rates? Or are the stratospheric aerosols treated in the

Interactive
comment
submodel RAD. Do the pressure levels of the submodel RAD and RAD_FUBRAD overlap? This needs to be explained more clearly.

**3   Submodel RAD**

1. Do the submodels RAD and RAD_FUBRAD overlap in height.

2. Is it possible for the authors to describe the differences between RAD_SHORT_v1 and RAD_SHORT_v2 in more detail.

**4   Sub-Submodel RAD_FUBRAD**

1. See comment B2 above

2. See comment B3 above

3. As the authors are giving a reference for the Ozone absorption cross sections in the Chappuis bands could they also specify where the other gaseous optical properties are coming from. I assume that they are either based on the HITRAN database or on GEISA.

**5   Submodel CLOUDOPT**

1. What are the options for cloud overlap?

2. What is the original reference for the ice-crystal optical properties. Are they based on A. Baran or P. Yang optical properties for example or something else.

**6 Spelling**

1. Section 2.1, line 16: replace USEd by used

2. Section 3.4, line 5 : ...  for cloud changes may arise in the context of direct anthropogenic...

3. Section 3.4, line 39: ... with almost arbitrary inputs.

---

## Referee Comment (RC2) · Anonymous Referee #2 · 16 Mar 2016

General comments

The paper presents a re-organized infrastructure of the radiation submodel within MESSy. So far the calculation of aerosol and cloud optical parameters as well as the radiative calculations themselves had been performed by one overall submodel, RAD4ALL. This submodel has now been split into 4 different and independent sub-models, which are based on the existing parameterizations. The new infrastructure allows for additional diagnostics and an online radiative forcing calculation. Overall I recommend the paper for publication. However, there are some sections where the paper needs clarification.

I really have a hard time to understand the different shortwave radiation schemes and

how they interact with each other. First of all, it's not clear to me why there are two versions of the shortwave scheme by Fouquart and Bonnel (1980). The statement "the combination of optical properties of different species are inconsistent" sounds to me as if there is a bug in RAD_SHORT_v1, so I don't see a reason to keep this version. Furthermore, I do not understand how RAD_FUBRAD and RAD_SHORT_v1/2 interact with each other. In the introduction it is written that RAD_FUBRAD works only in the stratosphere and mesosphere, so I guess the troposphere is handled by RAD_SHORT_v1/2. How is this done technically? And why is RAD_SHORT_CMN needed in addition to RAD_FUBRAD? Figure 1 is also not very helpful in this respect. So I think Sect. 2.1 would benefit from some further explanations. My overall impression is that there is still some space for improving the code structure in the shortwave part of the radiation.

According to the paper and the supplement it is possible to call the submodels AEROPT and CLOUDOPT several times for diagnostic purposes, as far as I understand. Is it possible to combine the various set-ups of AEROPT and CLOUDOPT for diagnostic calls within one model simulation?

Sections 2.6, 3.2, 3.3 and 3.4 provide some nice examples how the various submodels and diagnostics can be applied. However, I think the interpretation of the results is sometimes a bit sloppy. I am aware that this is a technical description paper, but nevertheless it would be good to see some more discussion, e.g. on the differences between SW_v1 and SW_v2 in Sect. 2.6 (details below).

Specific comments

- Introduction: The introduction clearly states the motivation behind the re-organization. I think it would be very helpful to clearly list the modifications and new diagnostics as well. For example, it is not clear to me whether it was already with RAD4ALL possible to call the radiation n times or with different aerosol or cloud set-ups.

- P3, L19: "the combination of ... is inconsistent..."

- Sect. 2.2: Spectral resolution of RAD_FUBRAD? On P4, L29 it is written that RAD_FUBRAD has 55 or 106 spectral bands in the UV-VIS band (250-690 nm). On P5, L8/9 it is written that the Chappuis band is resolved by either 1 band in the original version of the module or by 6 or 57 bands in the version of Kunze et al. (2014). So I assume that the overall 55 or 106 bands refer to the version of Kunze et al. (2014). How many bands in total does the original version then have? I am a bit confused about the various spectral resolutions. Please clarify.

- P5, L13: Shine and Rickaby (1989)

- P6, top: Please provide a reference for the CCMI stratospheric and volcanic aerosol data set.

- P7, L5: What is the purpose of several calls to the submodel AEROPT with different settings simultaneously?

- P8, L5: I have a general question on the spectral resolution of the shortwave scheme. Here you mention the 4 bands of the standard ECHAM5 shortwave radiation scheme. Cagnazzo et al. (2007) increased the number of shortwave bands from 4 to 6 for the middle atmosphere version of ECHAM5. Is this version of the shortwave scheme also available within the MESSy radiation code?

- P9, L15: It would be helpful to mention that the first call of the radiative calculation provides the temperature feedback. This information is only given in the supplement material, but I think it would be helpful to mention it in the main part of the paper as well.

- P9, L21: The statement that the calculated volcanic heating rates are comparable to Stenchikov et al. (1998) needs some more discussion. How comparable is the CCMI data set to Stenchikov et al. (1998) in terms of aerosol distribution and optical properties?

- P9, L23-25: Again I am a bit confused about the two versions of RAD_SHORT...

Which heating rates are more reliable, SW_v1 or SW_v2? Where do the differences between SW_v1 and SW_v2 come from and in which sense are they in agreement with Thomas (2008)?

- P11, L27: lead to -> led to ?

- P12, L11: For a better comparability - how large is the estimate by Stevenson et al. (2013)? Please provide the value.

- P13, L12: Could you provide the date of the next official MESSy release?

- Figure 1: What is the meaning of the different colors of the arrows? What does the dashed blue arrow to RAD_FUBRAF_E5 mean? The communication among the different shortwave routines is not absolutely clear to me (see major comment above).

- Table 1, caption: variated -> varied

- Table 2: net adjusted RF contrail: space missing -> 0.113 (0.113)

Specific comments on the supplement

- Is the description of the various namelists complete or is there only a subset of namelist parameters described?

- P4: "The namelist entry r_inp(m,n) then contains..."

- P4: "...decaying with elevation..." -> "...decaying with altitude..."

- P4: #vgrad: How do you specify the vertical gradient of the GHG VMR?

- P5: CTRL_FUBRAD, nbands=49: Why is this option still included if there are known bugs/shortcomings related to that specific spectral resolution?

- P5: Could you please briefly say what the acronym VISO means, for non-MESSy-experts?

- P8: CLOUDOPT, NCALL: Which call of CLOUDOPT is used for the radiative calculations, again the first call? Or can the n calls of CLOUDOPT be combined with n calls of the radiation scheme?

---

## Author Comment (AC1) · 10 May 2016

**Reply to Review 1**

We summarize our answers to the questions of review 1. Moreover the manuscript is changed taking into account the questions and comments (the changed manuscript is attached to the reply of review2).

**Comment (Abstract):** Could the author specify that the ECHAM general circulation model (GCM) has been developed by the Max-Planck Institut fuer Meteorologie for clarity purposes.

→ *Done, however not in the abstract. It is now mentioned in the introduction.*

**Question 1 (Introduction):** The authors are mentioning that the long-wave spectrum is divided into 16 bands ranging from 3.33-1000 microns. Could the authors please provide the same information for the UV-Vis and NIR. In section 2.2 it mentions that the UV-Vis band ranges from 250-690nm and therefore I would assume that the NIR is covered by three bands which range from 0.69 micron to 3.3 microns. But I am not sure that this is the case. This needs to be made clearer.

→ *The UV-VIS band ranges from 0.25 to 0.69µm and the NIR band from 0.69-4.00µm. Values are now mentioned in the manuscript .*

**Question 2 (Introduction):** The paper mentions that Rayleigh-Scattering is not considered in the submodel RAD_FUBRAD which looks at the UV-Vis regions at pressure levels below 70hPA, i.e. in the stratosphere and mesosphere. It seems to me that in the stratosphere and mesosphere Rayleigh-Scattering would be one of the most important radiative mechanism. Do the authors mean that the short-wave heating rates are not affected much by Rayleigh-Scattering? Or do the submodels RAD and RAD_FUBRAD overlap in pressure levels so that stratospheric and mesospheric Rayleigh-Scattering is treated in RAD. This needs to be clarified.

→ *Yes, FUBRad neglects Rayleigh scattering. RAD_FUBRAD and RAD do not overlap in pressure levels (if FUBRAD is switched on): FUBRAD replaces RAD for VIS-UV between TOA and 70 hPa. As mentioned in chapter 2.2 Rayleigh scattering is parametrized in the Chappuis und Huggins bands by a reflecting layer in the lower atmosphere. According to e.g., Strobel (1978), it is of sufficient accuracy for applications in MA GCMs (see also Interactive comment on Atmos. Chem. Phys. Discuss., 7, 45, 2007). Now clarified in chapter 2.2.*

**Question 3 (Introduction):** The paper also states that the submodel RAD_FUBRAD does not consider scattering by aerosols and clouds, although stratospheric aerosols are known to have an important radiative effect. Again, do the authors mean that they do not have a large effect on the heating rates? Or are the stratospheric aerosols treated in the submodel RAD. Do the pressure levels of the submodel RAD and RAD_FUBRAD overlap? This needs to be explained more clearly.

→ *Direct aerosol and cloud effects are not considered in FUBRAD. However, the reflection of UV-VIS on clouds and aerosols is considered in the upward flux, as mentioned in answer 2. Now clarified in chapter 1 and 2.2. Moreover, the effect of missing scattering on aerosols can be seen in figure 2 (compare RAD_SHORT_V1 and RAD_SHORT_V2, with and without FUBRAD respectively), not showing a big difference (up to maximal 10 %). Now clarified in chapter 2.2.*

**Question 1 (Submodel RAD)** Do the submodels RAD and RAD_FUBRAD overlap in height?

→ *RAD_FUBRAD and RAD do not overlap with height. If FUBRAD is switched on, shortwave radiation fluxes due to ozone and oxygen absorption are calculated at pressures equal or lower than 70hPa in the UV-Vis with FUBRAD (replacing the shortwave radiation scheme of Fouquart and Bonnel used in RAD). At pressures higher than 70 hPa the UV-Vis shortwave radiation fluxes are calculated by RAD_SHORT_V1 in one spectral interval as in the original ECHAM5 code, or modified as in RAD_SHORT_V2.*

**Question 2 (Submodel RAD)** Is it possible for the authors to describe the differences between RAD_SHORT_v1 and RAD_SHORT_v2 in more detail.

*→ Of course we can, a more detailed description is now given in section 2.1.*

**Question 1 (Sub-Submodel RAD_FUBRAD)**

*→* See answer above.

**Question 2 (Sub-Submodel RAD_FUBRAD)**

*→* See answer above.

**Question 3 (Sub-Submodel RAD_FUBRAD)** As the authors are giving a reference for the Ozone absorption cross sections in the Chappuis bands could they also specify where the other gaseous optical properties are coming from. I assume that they are either based on the HITRAN database or on GEISA.

*→ As the absorption cross sections are described in Nissen et al. (2007), they are not explicitly mentioned in the actual manuscript: Temperature-independent absorption cross sections are taken from Molina and Molina (1986) where available (206–347 nm) and from WMO (1986) between 347–362nm. For Lyman-α line the parametrized effective cross sections are depending on the $O_2$ slant column as suggested by Chabrillat and Kockarts (1997).*

**Question 1 (Submodel CLOUDOPT)** What are the options for cloud overlap?

*→ Here we used maximum random overlap in agreement with the ECHAM5 treatment (for details see Roeckner et al. 2003). The possible cloud overlap assumptions in radiation computation of EMAC are maximum-random overlap (default), maximum overlap and random overlap.*

**Question 2 (Submodel CLOUDOPT)** What is the original reference for the ice-crystal optical properties. Are they based on A. Baran or P. Yang optical properties for example or something else.

*→ The specific relations for the solar spectral bands are given in Rockel et al (1991) and are based on Mie calculation, a specific correction for the asymmetry factor is applied to account for non-sphericity of ice crystals (Roeckner et al, 2003). Mass absorption coefficients for liquid and ice clouds are parametrized as described by Roeckner et al. (2003) based on classical approaches from Stephens et al. 1990 and Ebert and Curry (1992). Text changed accordingly.*

**Spelling**

*→ Corrected.*

**References:**

Rockel, B., Raschke, E. and Weyres, B. (1991): A parameterization of broad band radiative transfer properties of water, ice and mixed clouds; Beitr. Phys. Atmosph., 64, 1-12.

Stephens, G. L., Tsay, S.-C., Stackhouse, P. W. and Flateau, P. J. (1990): The relevance of the microphysical and radiative properties of cirrus clouds to climate and climate feedback . J. Atmos. Sci., 47, 1742–1753.

Ebert, E. and Curry, J. A. (1992): A parameterization of cirrus cloud optical properties for climate models. J. Geophys. Res., 97, 3831–3836.

Roeckner, E., Bäuml, G., Bonaventura, L., Brokopf, R., Esch, M., Giorgetta, M., Hagemann, S., Kirchner, I., Kornblueh, L., Manzini, E., Rhodin, A., Schlese, U., Schulzweida, U. and Tompkins, A. 2003: The atmospheric general circulation model ECHAM5. Part I: Model description. Max Planck Institute for Meteorology Rep. 349, 127 pp.

Strobel (1978): Parameterization of the atmospheric heating rate from 15 to 120 km due to O2 and O3 absorption of solar radiation, Journal of Geophysical Research: Oceans, J. Geophys. Res., 83

Molina, L. T. and Molina, M.J. (1986): Absolute absorption cross sections of ozone in the 185- to 350-nm wavelength range, J. Geophys. Res., 91(D13), 14501–14508, doi:10.1029/JD091iD13p14501.

Chabrillat, S. and Kockarts, G.: Simple parameterization of the absorption of the solar Lyman–alpha line, Geophys. Res. Lett., 24, 815, 2659–2662, 1997.

World Meteorological Organization: Atmospheric ozone 1985, Global Ozone Res. Monit. Proj. Rep. 16/1, Geneva, 1986.

---

## Author Comment (AC2)

**Reply to Review 2**

We summarize our answers to the questions of referee #2. We agree with referee #2 that some sections of the manuscript, mainly section 2.1 need clarification. So we explain in more detail the two versions of the shortwave radiation scheme and also the interaction of FUBRAD and RAD_SHORT is clarified (see changed manuscript, attached to this reply; changes are highlighted). The manuscript is changed accordingly (wherever applicable).

**General Questions**

1) Difference of the two versions of the shortwave radiation schemes

*In **RAD_SHORT _V1** simplified assumptions for low aerosol loadings in the clear sky conditions are considered. For efficiency reasons, the effects of multiple reflection and the interactions between aerosol scattering and gaseous absorption were neglected (Thomas, 2008). The assumptions in RAD_SHORT _V1 are not valid for high aerosol loadings after volcanic eruptions. Thus, in **RAD_SHORT_V2** modifications were made in the model to include these effects, showing that the multiple reflection effect is a dominant effect for scattering particles (Thomas, 2008). Thus, the version RAD_SHORT _V2 is the improved shortwave version. As in the MESSy philosophy several different implementations of processes and diagnostics can coexist in the same model code, RAD_SHORT _V1 coexists, besides the more reliable version RAD_SHORT _V2. Advantage of this coexistence is e.g. the comparison of the two radiation schemes or to recalculation of older setups. So within the MESSy framework it would be also possible to implement the shortwave radiation scheme of Cagnazzo et al. (2007) as an additional alternative to RAD_SHORT _V1 and RAD_SHORT_V2.*

2) Interaction RAD_FUBRAD and RAD_SHORT

*RAD_FUBRAD is a sub-Submodel to RAD, which increases the resolution in the UV-Vis part of the solar spectrum. Yes, FUBRAD works only in the stratosphere and mesosphere. If FUBRAD is switched on, shortwave radiation fluxes due to ozone and oxygen are calculated at pressures equal or lower than 70hPa in the UV-Vis with FUBRAD (replacing the shortwave radiation scheme of Fouquart and Bonnel). At pressures higher than 70hPa in the UV-Vis shortwave radiation fluxes are calculated in one spectral interval by either RAD_SHORT_V1 or RAD_SHORT_V2. Thus, in regard to content RAD and FUBRAD are clearly separated. Technically there is some overlap between RAD_FUBRAD and RAD in the SMCL in the subroutine rad_sw_SW1S of RAD_SHORT _V1/V2 (more details are given in the text ).*

RAD_SHORT_CMN is not concerned with FUBRAD: The RAD_SHORT_CMN module contains definitions and an initialization subroutine, which are commonly used in RAD_SHORT_v1 and RAD_SHORT_v2, respectively (as mentioned in chapter 2.1). RAD_FUBRAD calculates heating rates in the middle atmosphere for the UV-Vis part of the solar spectrum. If the sub–submodel FUBRAD is switched on, RAD_FUBRAD is called from the shortwave calculation RAD_SHORT_v1 or RAD_SHORT_v2.

3) several calls of AEROPT and CLOUDOPT

*Yes it is possible to combine various setups of AEROPT and CLOUDOPT for diagnostic calls within one simulation. AEROPT and CLOUDOPT can be independently called several times.*
* * *
**Specific comments of Review 2:**

Introduction: The introduction clearly states the motivation behind the re-organization. I think it would be very helpful to clearly list the modifications and new diagnostics as well. For example, it is not clear to me whether it was already with RAD4ALL possible to call the radiation n times or with different aerosol or cloud set-ups.

*→ done! Besides the new structure of the radiation with the resulting independent submodels RAD, ORBIT, AEROPT, and CLOUDOPT, the most important modifications and new diagnostics are listed here:*

- *In RAD the import of external variables needed for radiation calculation (e.g. prescribed climatologies) is now done via the module IMPORT (data import from external files, Kerkweg and Jöckel, 2015).*
- *Within the submodel RAD a new important diagnostic feature is the calculation of radiative forcing by diagnostically calling the radiation routines several times.*
- *AEROPT can be called several times with different settings for the required aerosol optical properties simultaneously. At the time being three options for getting the aerosol optical properties are possible (Tanre climatology, offline input via IMPORT or online calculation).*
- *CLOUDOPT can be called several times and cloud optical properties of cloud coverages and cloud perturbations can be calculated individually.*
- *Updated version of FUBRAD with increased spectral resolution.*

P3, L19: "the combination of . . . is inconsistent. . ."

→ *As text has changed, this is obsolete.*

Sect. 2.2: Spectral resolution of RAD_FUBRAD? On P4, L29 it is written that RAD_FUBRAD has 55 or 106 spectral bands in the UV-VIS band (250-690 nm). On P5, L8/9 it is written that the Chappuis band is resolved by either 1 band in the original version of the module or by 6 or 57 bands in the version of Kunze et al. (2014). So I assume that the overall 55 or 106 bands refer to the version of Kunze et al. (2014). How many bands in total does the original version then have? I am a bit confused about the various spectral resolutions. Please clarify.

→ *Possible are 55 (default), 106, and 49 spectral bands. The old (original) version has 49 spectral bands, however this version is not recommended as it leads to an inconsistent flux profile and misleading flux diagnostics. The 49 spectral bands of the old version are now mentioned in the text.*

P5, L13: Shine and Rickaby (1989)

→ *As text has changed, this is obsolete.*

P6, top: Please provide a reference for the CCMI stratospheric and volcanic aerosol

→ *Done. For CCMI Input data there is no reference in peer reviewed literature. They can be found under the link* [ftp://iacftp.ethz.ch/pub_read/luo/ccmi](ftp://iacftp.ethz.ch/pub_read/luo/ccmi)*. And there is a "Release note" there.*

P7, L5: What is the purpose of several calls to the submodel AEROPT with different settings simultaneously?

→ *This is the requirement for the (optional) multiple diagnostic calls of the radiation routines with different aerosol properties: AEROPT can be diagnostically called with different settings for aerosol optical properties simultaneously, e.g . with different approaches (internal or external mixture) or with different parametrisations (TANRE vs explicit calculations). Then, by calling the radiation routines with different aerosol optical properties, their radiative forcing can then be determined, including sensitivities with respect to the stetting.*

P8, L5: I have a general question on the spectral resolution of the shortwave scheme. Here you mention the 4 bands of the standard ECHAM5 shortwave radiation scheme. Cagnazzo et al. (2007) increased the number of shortwave bands from 4 to 6 for the middle atmosphere version of ECHAM5. Is this version of the shortwave scheme also available within the MESSy radiation code?

→ *The shortwave radiation scheme of Cagnazzo et al. (2007) is not included in EMAC yet, however with the new, MESSY conform infrastructure of the radiation code, it should be technically easy to extend the MESSy code by an additional shortwave radiation scheme (besides the existing versions RAD_short_v1/v2).*

P9, L15: It would be helpful to mention that the first call of the radiative calculation provides the temperature feedback. This information is only given in the supplement material, but I think it would be helpful to mention it in the main part of the paper as well.

*→ Done in chapter 2.1.*

P9, L21: The statement that the calculated volcanic heating rates are comparable to Stenchikov et al. (1998) needs some more discussion.

*→ Done. For August 1991 the heating rates of our model are in structural good agreement with Stenchikov et al. (1998), showing the maximum between 0-10°S at 20 hPa, however values are higher (up to a maximum of 0.9 K/d ) in our model. Also January 1992 shows structural good agreement with Stenchikov et al. (1998) (zonally averaged pictures of August 1991 and January 1992 are not shown).*

How comparable is the CCMI data set to Stenchikov et al. (1998) in terms of aerosol distribution and optical properties?

*→ We did not compare the two data sets explicitly, because this is beyond the scope of this paper. The data set of Stenchikov et al. (1998) is a spectral-, space-, and time-dependent set of aerosol parameters for 2 years after the Pinatubo eruption using a combination of SAGE II aerosol extinctions and UARS-retrieved effective radii, supported by SAM II, AVHRR, lidar and balloon observations. The CCMI data set is primarily based on the SAGE series of measurements, but has been extended in time using CALIPSO and GOMOS, and into the past using volcanic records and simple modeling and now spans 1960 to 2013.*

P9, L23-25: Again I am a bit confused about the two versions of RAD_SHORT. . . Which heating rates are more reliable, SW_v1 or SW_v2? Where do the differences between SW_v1 and SW_v2 come from and in which sense are they in agreement with Thomas (2008)?

*→ The heating rates of SW_v2 are more reliable, as the effects of multiple reflection and the interactions between aerosol scattering and gaseous absorption (which are important for high aerosol loadings after volcanic eruptions) are neglected in SW_v1 (see Thomas, 2008). That is why, in agreement with Thomas et al. (2008), SW heating rates are overestimated due to the simplified assumptions for low aerosol loadings in the clear sky conditions, which are not valid for high aerosol loadings after volcanic eruptions (see Figure 2). Text changed accordingly.*

P11, L27: lead to -> led to ?
*→done*

P12, L11: For a better comparability - how large is the estimate by Stevenson et al. (2013)? Please provide the value.

*→ Sorry, the citation was wrong. The correct citation is Myhre et al. 2013, their table 8.3. The corresponding value of -0.02 W/m2 is now given in the text.*

P13, L12: Could you provide the date of the next official MESSy release?

*→ There is no regular date of the official MESSy releases. However, all modifications and new diagnostics mentioned in this paper are implemented in the versions 2.51 and 2.52, which are available now.*

Figure 1: What is the meaning of the different colors of the arrows? What does the dashed blue arrow to RAD_FUBRAD_E5 mean? The communication among the different shortwave routines is not absolutely clear to me (see major comment above).

*→The blue arrows indicate the input to RAD and RAD_FUBRAD (dashed) via the channel infrastructure and the red arrows indicate the trigger, passed from RAD to ORBIT. The black arrows indicate the dependencies of the Fortran95 modules through Fortran USE statements. The direction of the arrows indicates where the different modules are used. For the communication between the different shortwave routines see answers above. Changed figure caption accordingly.*

Table 1, caption: variated -> varied
→ *corrected*

Table 2: net adjusted RF contrail: space missing -> 0.113 (0.113)
→ *corrected*

**Specific comments on the supplement**

Is the description of the various namelists complete or is there only a subset of namelist parameters described?
→ *Namelists are complete.*

P4: "The namelist entry r_inp(m,n) then contains. . ."
→ *Corrected.*

P4: ". . .decaying with elevation. . ." -> ". . .decaying with altitude. . ."
→ *Corrected.*

P4: #vgrad: How do you specify the vertical gradient of the GHG VMR?

→ *The formula of calculating the vertical gradient of the GHG volume mixing ratio is now given in the text.*

P5: CTRL_FUBRAD, nbands=49: Why is this option still included if there are known bugs/shortcomings related to that specific spectral resolution?

→ *MESSy philosophy: several different implementations of processes and diagnostics can coexist in the same model code (for e.g. for comparison or recalculation).*

P5: Could you please briefly say what the acronym VISO means, for non-MESSy experts?

→ *Yes. The diagnostic submodel, VISO, serves two purposes. First, it is used to diagnose vertically layered, 2-D iso-surfaces in 3-D scalar fields in Eulerian (grid-point) representation. The second application of VISO is for mapping 3-D scalar fields in Eulerian (grid-point) representation on surfaces defined by a level index (and optionally by a fraction of the box), as for instance an iso-surface defined by the same submodel ( Jöckel et al., 2010, Sect. 5.1)*

P8: CLOUDOPT, NCALL: Which call of CLOUDOPT is used for the radiative calculations, again the first call? Or can the n calls of CLOUDOPT be combined with n calls of the radiation scheme?

→ *Yes, the n calls of CLOUDOPT can be combined with n calls of the radiation scheme.*

**References:**

Myhre, G., D. Shindell, F.-M. Bréon, W. Collins, J. Fuglestvedt, J. Huang, D. Koch, J.-F. Lamarque, D. Lee, B. Mendoza, T. Nakajima, A. Robock, G. Stephens, T. Takemura and H. Zhang, 2013: Anthropogenic and Natural Radiative Forc-ing. In: Climate Change 2013: The Physical Science Basis. Contribution of Working Group I to the Fifth Assessment Report of the Intergovernmental Panel on Climate Change [Stocker, T.F., D. Qin, G.-K. Plattner, M. Tignor, S.K. Allen, J. Boschung, A. Nauels, Y. Xia, V. Bex and P.M. Midgley (eds.)]. Cambridge University Press, Cambridge, United Kingdom and New York, NY, USA.

[revised manuscript text omitted]
 | 0.48 (0.27) | 0.06 (-0.16) | 0.115 (0.109) | 0.45 (0.45) | -0.01 (-0.01) | 0.113 (0.113) |

---

## Author Response (AR1)

Oberpfaffenhofen, 10th Mai 2016

Dear Dr. Olaf Morgenstern,

We submit the revised version of our manuscript (doi:10.5194/gmd-2015-277):

 **"A new radiation infrastructure for the Modular Earth Submodel System (MESSy, based on version 2.51)"**

**by S. Dietmüller, P. Jöckel, H. Tost, M. Kunze, C. Gellhorn, S. Brinkop, C. Frömmming, M. Ponter, B. Steil, A. Lauer and J. Hendricks**

We have carefully considered all points brought up by the two reviewers. The responses to the Referees are now uploaded and also attached to this letter. Furthermore we have highlighted all changes in the manuscript and also attached them to this letter.

We thank the reviewers for the constructive comments, which made the manuscript clearer and more informative.

Sincerely,

Dr. Simone Dietmüller (on behalf of all co-authors)

**Reply to Review 1**

We summarize our answers to the questions of review 1. Moreover the manuscript is changed taking into account the questions and comments (the changed manuscript is attached to the reply of review2).

**Comment (Abstract):** Could the author specify that the ECHAM general circulation model (GCM) has been developed by the Max-Planck Institut fuer Meteorologie for clarity purposes.

→ *Done, however not in the abstract. It is now mentioned in the introduction.*

**Question 1 (Introduction):** The authors are mentioning that the long-wave spectrum is divided into 16 bands ranging from 3.33-1000 microns. Could the authors please provide the same information for the UV-Vis and NIR. In section 2.2 it mentions that the UV-Vis band ranges from 250-690nm and therefore I would assume that the NIR is covered by three bands which range from 0.69 micron to 3.3 microns. But I am not sure that this is the case. This needs to be made clearer.

→ *The UV-VIS band ranges from 0.25 to 0.69µm and the NIR band from 0.69-4.00µm. Values are now mentioned in the manuscript .*

**Question 2 (Introduction):** The paper mentions that Rayleigh-Scattering is not considered in the submodel RAD_FUBRAD which looks at the UV-Vis regions at pressure levels below 70hPA, i.e. in the stratosphere and mesosphere. It seems to me that in the stratosphere and mesosphere Rayleigh-Scattering would be one of the most important radiative mechanism. Do the authors mean that the short-wave heating rates are not affected much by Rayleigh-Scattering? Or do the submodels RAD and RAD_FUBRAD overlap in pressure levels so that stratospheric and mesospheric Rayleigh-Scattering is treated in RAD. This needs to be clarified.

→ *Yes, FUBRad neglects Rayleigh scattering. RAD_FUBRAD and RAD do not overlap in pressure levels (if FUBRAD is switched on): FUBRAD replaces RAD for VIS-UV between TOA and 70 hPa. As mentioned in chapter 2.2 Rayleigh scattering is parametrized in the Chappuis und Huggins bands by a reflecting layer in the lower atmosphere. According to e.g., Strobel (1978), it is of sufficient accuracy for applications in MA GCMs (see also Interactive comment on Atmos. Chem. Phys. Discuss., 7, 45, 2007). Now clarified in chapter 2.2.*

**Question 3 (Introduction):**The paper also states that the submodel RAD_FUBRAD does not consider scattering by aerosols and clouds, although stratospheric aerosols are known to have an important radiative effect. Again, do the authors mean that they do not have a large effect on the heating rates? Or are the stratospheric aerosols treated in the submodel RAD. Do the pressure levels of the submodel RAD and RAD_FUBRAD overlap? This needs to be explained more clearly.

→ *Direct aerosol and cloud effects are not considered in FUBRAD. However, the reflection of UV-VIS on clouds and aerosols is considered in the upward flux, as mentioned in answer 2. Now clarified in chapter 1 and 2.2. Moreover, the effect of missing scattering on aerosols can be seen in figure 2 (compare RAD_SHORT_V1 and RAD_SHORT_V2, with and without FUBRAD respectively), not showing a big difference (up to maximal 10 %). Now clarified in chapter 2.2.*

**Question 1 (Submodel RAD)** Do the submodels RAD and RAD_FUBRAD overlap in height?

→ *RAD_FUBRAD and RAD do not overlap with height. If FUBRAD is switched on, shortwave radiation fluxes due to ozone and oxygen absorption are calculated at pressures equal or lower than 70hPa in the UV-Vis with FUBRAD (replacing the shortwave radiation scheme of Fouquart and Bonnel used in RAD). At pressures higher than 70 hPa the UV-Vis shortwave radiation fluxes are calculated by RAD_SHORT_V1 in one spectral interval as in the original ECHAM5 code, or modified as in RAD_SHORT_V2.*

**Question 2 (Submodel RAD)** Is it possible for the authors to describe the differences between RAD_SHORT_v1 and RAD_SHORT_v2 in more detail.

→ *Of course we can, a more detailed description is now given in section 2.1.*

**Question 1 (Sub-Submodel RAD_FUBRAD)**

→ See answer above.

**Question 2 (Sub-Submodel RAD_FUBRAD)**

→ See answer above.

**Question 3 (Sub-Submodel RAD_FUBRAD)** As the authors are giving a reference for the Ozone absorption cross sections in the Chappuis bands could they also specify where the other gaseous optical properties are coming from. I assume that they are either based on the HITRAN database or on GEISA.

→ *As the absorption cross sections are described in Nissen et al. (2007), they are not explicitly mentioned in the actual manuscript: Temperature-independent absorption cross sections are taken from Molina and Molina (1986) where available (206–347 nm) and from WMO (1986) between 347–362nm. For Lyman-α line the parametrized effective cross sections are depending on the $O_2$ slant column as suggested by Chabrillat and Kockarts (1997).*

**Question 1 (Submodel CLOUDOPT)** What are the options for cloud overlap?

→ *Here we used maximum random overlap in agreement with the ECHAM5 treatment (for details see Roeckner et al. 2003). The possible cloud overlap assumptions in radiation computation of EMAC are maximum-random overlap (default), maximum overlap and random overlap.*

**Question 2 (Submodel CLOUDOPT)** What is the original reference for the ice-crystal optical properties. Are they based on A. Baran or P. Yang optical properties for example or something else.

→ *The specific relations for the solar spectral bands are given in Rockel et al (1991) and are based on Mie calculation, a specific correction for the asymmetry factor is applied to account for non-sphericity of ice crystals (Roeckner et al, 2003). Mass absorption coefficients for liquid and ice clouds are parametrized as described by Roeckner et al. (2003) based on classical approaches from Stephens et al. 1990 and Ebert and Curry (1992). Text changed accordingly.*

**Spelling**

→ *Corrected.*

P13, L12: Could you provide the date of the next official MESSy release?

*→ There is no regular date of the official MESSy releases. However, all modifications and new diagnostics mentioned in this paper are implemented in the versions 2.51 and 2.52, which are available now.*

Figure 1: What is the meaning of the different colors of the arrows? What does the dashed blue arrow to RAD_FUBRAD_E5 mean? The communication among the different shortwave routines is not absolutely clear to me (see major comment above).

*→The blue arrows indicate the input to RAD and RAD_FUBRAD (dashed) via the channel infrastructure and the red arrows indicate the trigger, passed from RAD to ORBIT. The black arrows indicate the dependencies of the Fortran95 modules through Fortran USE statements. The direction of the arrows indicates where the different modules are used. For the communication between the different shortwave routines see answers above. Changed figure caption accordingly.*

Table 1, caption: variated -> varied
→ *corrected*

Table 2: net adjusted RF contrail: space missing -> 0.113 (0.113)
→ *corrected*

**Specific comments on the supplement**

Is the description of the various namelists complete or is there only a subset of namelist parameters described?
→ *Namelists are complete.*

P4: "The namelist entry r_inp(m,n) then contains. . ."
→ *Corrected.*

P4: ". . .decaying with elevation. . ." -> ". . .decaying with altitude. . ."
→ *Corrected.*

P4: #vgrad: How do you specify the vertical gradient of the GHG VMR?

→ *The formula of calculating the vertical gradient of the GHG volume mixing ratio is now given in the text.*

P5: CTRL_FUBRAD, nbands=49: Why is this option still included if there are known bugs/shortcomings related to that specific spectral resolution?

→ *MESSy philosophy: several different implementations of processes and diagnostics can coexist in the same model code (for e.g. for comparison or recalculation).*

P5: Could you please briefly say what the acronym VISO means, for non-MESSy experts?

→ *Yes. The diagnostic submodel, VISO, serves two purposes. First, it is used to diagnose vertically layered, 2-D iso-surfaces in 3-D scalar fields in Eulerian (grid-point) representation. The second application of VISO is for mapping 3-D scalar fields in Eulerian (grid-point) representation on surfaces defined by a level index (and optionally by a fraction of the box), as for instance an iso-surface defined by the same submodel ( Jöckel et al., 2010, Sect. 5.1)*

P8: CLOUDOPT, NCALL: Which call of CLOUDOPT is used for the radiative calculations, again the first call? Or can the n calls of CLOUDOPT be combined with n calls of the radiation scheme?

→ *Yes, the n calls of CLOUDOPT can be combined with n calls of the radiation scheme.*

**References:**

Myhre, G., D. Shindell, F.-M. Bréon, W. Collins, J. Fuglestvedt, J. Huang, D. Koch, J.-F. Lamarque, D. Lee, B. Mendoza, T. Nakajima, A. Robock, G. Stephens, T. Takemura and H. Zhang, 2013: Anthropogenic and Natural Radiative Forc-ing. In: Climate Change 2013: The Physical Science Basis. Contribution of Working Group I to the Fifth Assessment Report of the Intergovernmental Panel on Climate Change [Stocker, T.F., D. Qin, G.-K. Plattner, M. Tignor, S.K. Allen, J. Boschung, A. Nauels, Y. Xia, V. Bex and P.M. Midgley (eds.)]. Cambridge University Press, Cambridge, United Kingdom and New York, NY, USA.

[revised manuscript text omitted]

$$0.5 * (vmr + c1 * vmr) * (1 - \frac{vmr + c1 * vmr}{vmr - c1 * vmr} * tanh\frac{log(\frac{p}{c2})}{c3}), \tag{1}$$

    with vmr (in mol/mol) from namelist. The constants c1, c2 and c3 differ for the different GHGs and are given in Table 1.
    Example: `r_inp(1,2) = '#vgrad', '``CH4=1.75E-06'`

Table 1: Constants c1, c2 and c3 for calculating the vertical gradient of the GHGs $CH_4$, $N_2O$, CFC-11 and CFC-12 (see equation 1).

|  | c1 | c2 | c3 |
|---|---|---|---|
| $CH_4$ | 0.125 | 683 | -1.43 |
| $N_2O$ | 0.012 | 1395 | -1.43 |
| CFC-11 | 0.0001 | 4159 | -0.73 |
| CFC-12 | 0.0001 | 3177.4 | -0.73 |

**2.2  CTRL_FUBRAD and CPL_FUBRAD namelist**

The namelist file *rad.nml* contains two namelists for the high resolution shortwave radiation scheme FUBRAD, `CTRL_FUBRAD` and `CPL_FUBRAD`:

- With the namelist parameter `solfac` in `CTRL_FUBRAD` the solar cycle condition can be set. It can vary between 0 and 1, where the value 0 indicates solar minimum and the value 1 solar maximum conditions, respectively. This parameter is obsolete, if `fubrad_solar` in `CPL_FUBRAD` is activated.

- The second parameter in `CTRL_FUBRAD`, `nbands`, sets the spectral resolution of FUBRAD. Possible are 55 (default), 106, and 49 bands. However the old version of 49 bands is not recommended, as it leads to an inconsistent flux profile and misleading flux diagnostics.

- In `CPL_FUBRAD` the channel object providing an external solar cycle time series can be chosen with `fubrad_solar`. If such a time series is, for instance, imported with the submodel IMPORT_TS (Kerkweg and Jöckel, 2015), the channel name (first string) is 'import_ts' followed by the channel object name (second string), e.g. 'solact', or 'solspec'. The time series may consist of either one parameter (the F10.7 cm flux), or of `nbands` + 1 parameters for spectrally resolved data. If `fubrad_solar` is commented (or empty), `solfac` in `CTRL_FUBRAD` is used instead.

**2.3  RAD CPL namelist settings for radiative forcing calculations**

The RAD submodel can be called several times with different radiation settings, as the calculation is fully diagnostic, besides the first call, which provides the temperature feedback. As mentioned above, `l_switch(n)` switches the $n$th radiation calculation. With this new feature in the radiation submodel RAD the user can calculate instantaneous and stratosphere adjusted radiative forcing during a model simulation. The namelist file *rad.nml* in Table 2 shows an example of one additional diagnostic call ($n = 2$) for calculating stratosphere adjusted radiative forcing of enhanced $CO_2$ mixing ratios (`r_inp(2,2) ='#const','CO2=416.E-6',`). In the second call, radiation is diagnostically calculated (without temperature feedback) for the changed $CO_2$ mixing ratios. The empty strings (see Table 2) for the second call are automatically replaced by the corresponding entries of the first call, such that only modifications need to be listed.

With the submodel VISO, which is mapping 3-D scalar fields in grid-point representation on surfaces defined by a level index (
[revised manuscript text omitted]